

# The influence of snow cover dynamics on gross primary productivity of cultivated land in Northeast China

Lue Li[1], Qian Yang[1,*], Meng Cui[2], Huanjun Liu[1], Xiaohua Hao[3], Yiyang Peng[2] and Junyi Chang[1]

5    [1] Northeast Institute of Geography and Agroecology, Chinese Academy of Sciences, Remote Sensing and Geographic Information Research Centre, Changchun, 130102, China

[2] Cultivated Land Quality, Monitoring and Protection center, Ministry of Agriculture and Rural Affairs, Beijing, 100125, China

[3] Northwest Institute of Eco-Environment and Resources, Chinese Academy of Sciences, Lanzhou, 10    730000, China

*Correspondence: Qian Yang, Email-address: yangqian@iga.ac.cn, Tel.: +86-131-0441-6860*

**Abstract.** Snow cover is a crucial factor influencing gross primary productivity (GPP), but the various regulatory mechanisms across different geographical zones in Northeast China remain unclear. This study comprehensively analyzed the dynamic changes in snow cover and GPP in **Northeast** China 15    from HY2001 to HY2020. Specifically, the study area was divided into six subregions to investigate the impact of snow cover on cropland GPP. The results revealed that the snow water equivalent (SWE) decreased in 63% of the croplands in Northeast China, while the snow cover duration (SCD) increased in 54% of the croplands. Additionally, delayed snow cover end dates (SCEDs) were observed in 61% of the croplands, with 74% showing significant increases in cropland GPP. In terms of cropland types, 20    SCD showed the strongest positive correlation with dry lands, while paddy fields were more sensitive to SCED variations. Geographically, the Changbai Mountain, Sanjiang Plain, and Khingan Ranges exhibited more pronounced GPP changes due to SCED. In contrast, the Liaohe Plain and Western Sand Area were predominantly affected by SWE, while the Songnen Plain showed greater sensitivity to SCD. These findings elucidate the critical role of snow cover in modulating cropland GPP variations across 25    different geographical zones, providing valuable insights into the influence of similar climatic conditions on cropland ecosystems.



Key words: snow cover, gross primary productivity, cropland, Northeast China

## 1 Introduction

Croplands are critical natural resources for ensuring food security, ecological stability, and economic sustainability. Under the pressure of intensifying soil erosion and climate change, understanding cropland gross primary productivity (GPP) variation trends and its environmental response mechanisms has become imperative for sustainable development and enhanced cropland conservation. GPP represents vegetation's photosynthetic carbon fixation capacity per unit time and serves as a key metric of carbon assimilation through photosynthesis (Beer et al., 2010; Sjöström et al., 2013). Cropland ecosystems play are pivotal role in terrestrial carbon cycling (Wang et al., 2022), where GPP directly governs crop growth dynamics, carbon sequestration potential, and agricultural productivity variations, making it an essential indicator of agroecosystem productivity. (Wagle et al., 2015). As a key component of terrestrial ecosystems, snow cover significantly affects the carbon cycle through ecosystem functioning modification. Under the trend of global warming in recent decades, significant changes in snow cover have been observed (Mudryk et al., 2020; Pulliainen et al., 2020), which subsequently influence vegetation dynamics and GPP through altered environmental conditions (Meredith et al., 2019).

Previous studies demonstrated that snow cover and its phenological changes regulate surface energy balance and hydrological cycles while directly affecting the growing season timing and photosynthetic efficiency in croplands, thereby modulating GPP. Winter snow water equivalent (SWE) and snow cover duration (SCD) largely determine soil moisture availability and thermal regimes (Blankinship and Hart, 2012). These regulatory effects prove particularly crucial during spring sowing periods, with lasting impacts on annual carbon uptake efficiency (Chen et al., 2019). However, the mechanisms underlying snow cover's influences on GPP exhibit significant heterogeneity across vegetation types and geographical contexts. Delayed snow cover end dates (SCEDs) could enhance early growing season GPP in dry land and grassland ecosystems while negatively affecting forest GPP (Wang et al., 2024). In relatively arid regions, snow cover's stronger positive hydrological effects on soil enhance GPP more significantly (Liu et al., 2023). Different snow cover indicators demonstrate varying modes of influence: SWE affects GPP through soil moisture and nitrogen dynamics modification, while a thick snow cover





protects root systems from winter mortality (Brooks et al., 2011; Knowles et al., 2017). Meanwhile, SCD decreases are linked to advanced vegetation phenology and subsequent productivity increases (Pulliainen et al., 2017). These effects are further modulated by climate change drivers including temperature rise and precipitation variability (Peng et al., 2010). GPP is a comprehensive indicator of the complex interactions among climatic, topographic, edaphic, botanical, and anthropogenic factors.

Currently, few systematically investigated the snow cover-cropland GPP relationship across distinct geographical zones while considering these multifaceted interactions.

Northeast China hosts a vital grain production base that is crucial for national food security. Due to a growing population and intensifying climate change, understanding regional GPP responses across geographical conditions has become increasingly urgent. This study integrates multi-source data,

including long-term remote sensing observations (2001 to 2020), climate records, snow cover indicators, and agricultural statistics, to systematically analyze the spatiotemporal snow cover variation patterns and their mechanistic impacts on cropland GPP in Northeast China. The objectives include (1) examining the spatiotemporal snow cover variations (e.g., SWE, SCD, and SCED), (2) elucidating the spatiotemporal heterogeneity of snow cover's effects on cropland GPP, and (3) exploring the

mechanisms underlying the regulatory roles of snow cover on cropland GPP.

## 2 Materials and methods

### 2.1 Study area

Northeast China is a high-latitude region (38°72' to 53°56'N, 115°52' to 135°09'E) comprising Heilongjiang, Jilin, and Liaoning provinces, along with the eastern four leagues of the Inner Mongolia

Autonomous Region. The region covers approximately 1.25 million $km^2$ and hosts 358,700 $km^2$ of cropland, accounting for 26.6% of China's total cultivated area (Wang et al., 2023a). The topography exhibits distinct regional differentiation, with mountainous peripheries on three sides and extensive plains in the interior. Six major geographical units exist in the region: the Songnen Plain, Sanjiang Plain, Liaohe Plain, Khingan Range, Khingan Range, and Changbai Mountain (Figure 1). These

geographical variations led to distinct climatic characteristics across these six sub-regions (Table 1).



The region features a temperate monsoon climate characterized by winter snowfall (Xue et al., 2022), low evaporation rates, and high humidity. The winters are cold and prolonged, particularly in higher latitudes. The annual effective accumulated temperature ranges from 2320 ° C to 3654 ° C, while annual precipitation varies between 238 mm and 1078 mm (Wang et al., 2023b). The average elevation stands at approximately 200 meters above sea level. As one of the country's largest seasonal snow cover regions, Northeast China is a crucial agricultural zone with the highest grain production nationwide (Ma et al., 2024). The predominant cropping systems allow a single annual harvest pattern. Therefore, investigating cropland ecosystem responses to snow cover variations is of great significance to regional agriculture.

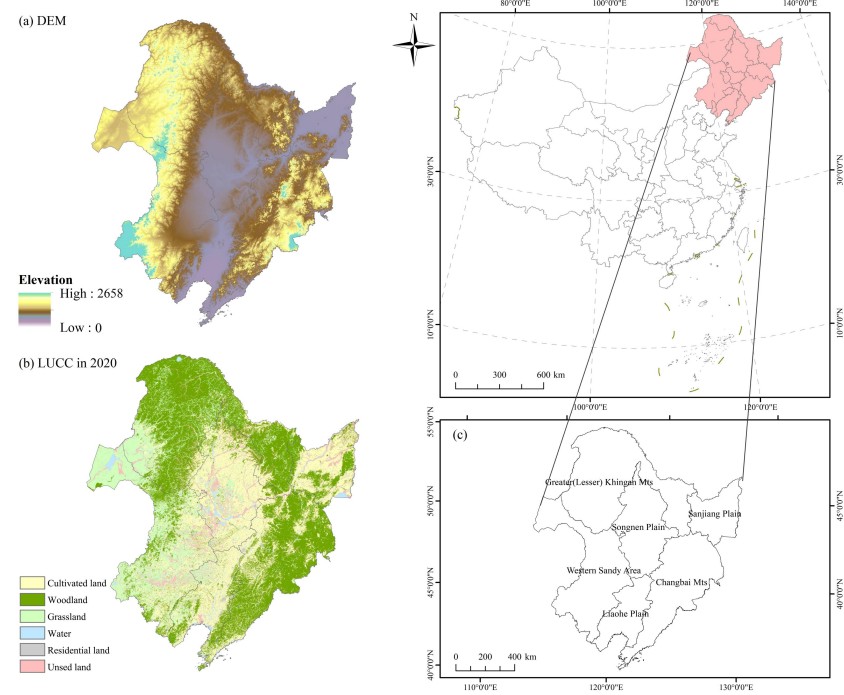

Figure 1. The overview of study area: (a) elevation; (b) land use types; (c) six geographic divisions.

Table 1 The details of six geographic divisions in Northeast China

| Geographical region | Abbre | Area (10^4×k m^2) | Elevation range (m) | Cultivated land (thousand km^2) | Accumulated temperature ≥ 10 ℃ | Precipitation (mm) |
|---|---|---|---|---|---|---|



| Songnen Plain | SN | 18.35 | 95~957 | 107.6 | 2706 | 400 ~ 650 |
| Sanjing Plain | SJ | 10.18 | 0~1030 | 66.8 | 2402 | 600 ~ 800 |
| Liaohe Plain | LH | 10.57 | 0~1215 | 35.3 | 3654 | 500 ~ 700 |
| Changbai Mount | CB | 24.64 | 0~2658 | 64.6 | 2857 | 800 ~ 1200 |
| Western Sand Area | WS | 26.14 | 115~2015 | 53.6 | 3262 | 200 ~ 400 |
| Daxing'an and Xiaoxing'an Mountain | XAL | 34.55 | 67~1079 | 30.4 | 2320 | 400 ~ 700 |

## 2.2 Materials

### 2.2.1 Snow cover products

Three snow cover indicators were utilized in this study: SWE, SCD, and SCED. The SWE data were obtained from the National Cryosphere Desert Data Center, specifically the 0.25° daily SWE fusion product for China (1980 to 2020) (Jiang et al., 2022). This dataset integrated the advantages of existing SWE data products with topographic and temporal covariates and was validated using ground observations from 647 monitoring stations. The validation results demonstrated correlation coefficients ($R^2$) of 0.77 and 0.70, with mean absolute errors (MAE) of 7.54 mm and 8.62 mm and root mean square errors (RMSE) of 12.29 mm and 13.73 mm, respectively.

The SCD and SCED data were derived from the MODIS-based Chinese Snow Phenology Dataset (2000 to 2020) (Zhao et al., 2022). which provides a spatial resolution of 500 m around China. The dataset has been rigorously validated against ground station observations, showing high accuracy. The $R^2$, RMSE, and MAE of SCD are 0.94, 12.09 days, and 7.60 days, respectively. The $R^2$, RMSE, and MAE of SCED are 0.56, 19.89 days, and 7.74 days, respectively.



### 2.2.2 GPP data

The MOD17A2H version 6 GPP data product (2001 to 2020) was adopted, (Running et al., 2021). This product provides 8-day composite data at 500 m spatial resolution, offering cumulative measurements

of vegetation photosynthetic activity. MODIS products have been extensively validated and widely adopted in terrestrial carbon cycle research (Endsley et al., 2023; Wang et al., 2017), Existing studies have validated the MOD17A2 GPP data product across various ecosystems in China, demonstrating a strong agreement with in-situ eddy covariance flux tower observations ($R^2 = 0.76$) (Zhu et al., 2016).

Version updates incorporate algorithm refinements and enhanced processing techniques, improving

accuracy and reliability across successive iterations. These validations warrant its suitability for scientific applications in quantifying terrestrial energy fluxes, carbon-water cycles, and vegetation biogeochemical dynamics. Its robustness enables effective applications in modeling ecosystem productivity and analyzing climate-vegetation interactions at regional scales.

### 2.2.3 Climate data

The precipitation, air temperature, and solar radiation data were employed to analyze the domain factors influencing snow cover. The monthly precipitation and temperature data were obtained from the 1 km-resolution monthly precipitation dataset (Peng, 2020) and the 1 km-resolution monthly mean temperature dataset for China (Peng, 2019). These datasets provide monthly records from 1901 to 2021 across China at a spatial resolution of 1 km, comprehensively covering various climatic variables. Both

datasets have been validated against 496 independent meteorological stations, demonstrating their high reliability and accuracy in representing regional climate patterns.

The solar radiation data were acquired from the ERA5-LAND hourly reanalysis product comprising continuous surface solar radiation estimates. The accuracy of the ERA5 solar radiation data has been extensively validated in multiple studies (Mihalevich et al., 2022; Muñoz-Sabater et al., 2021).

Comprehensive evaluation under diverse environmental conditions demonstrated that the ERA5-LAND product accurately represents actual solar radiation patterns, with high suitability for various ecological





and climatological applications. Integrating these high-quality climate datasets enables robust analysis of climate-vegetation interactions and ecosystem dynamics at regional scales.

All remote sensing data used in this study were standardized to the temporal range of HY2001 to
HY2020 and resampled to a uniform spatial resolution of 0.05° × 0.05° using the nearest neighbor algorithm. This standardization enables consistent spatiotemporal comparisons across different datasets, and the selected resolution balances computational efficiency and spatial detail. The resampling approach preserves original data values and minimizes interpolation artifacts to facilitate subsequent pixel-by-pixel analysis of raster data.

**2.2.4 Soil data**

Soil temperature (ST) and soil moisture (SM) data were obtained from the Famine Early Warning Systems Network (FEWS NET) Land Data Assimilation System (FLDAS) (McNally, 2018). The FLDAS data are generated using the Noah Land Surface Model (LSM) version 3.6.1, with a spatial resolution of 0.1° × 0.1° and monthly temporal resolution, providing 28 surface variables from 1982 to
present. This comprehensive dataset includes the 0 to 10 cm SM and temperature data used in this study, which are particularly relevant for analyzing vegetation dynamics and ecosystem processes.

The FLDAS data have been validated against multiple in-situ soil observation networks and demonstrated superior accuracy to the Global Land Data Assimilation System (GLDAS) (Li et al., 2021). The validation involved extensive comparisons with ground-based measurements, confirming
the reliability of FLDAS outputs for soil parameter estimation. This high-quality dataset enables robust analysis of soil-vegetation-atmosphere interactions and supports various applications in ecological modeling and climate studies.

**2.2.5 Land use type**

The land use data were obtained from the 1 km-resolution China Land Use Dataset (1980 to 2020) (Xu
et al., 2018). This dataset employs a two-level classification scheme, with the first comprising 6 major categories (forest, grassland, cropland, water bodies, built-up areas, and barren land) and the second containing 23 subcategories. To better characterize the effects of snow cover on cropland GPP, we





utilized the second-level classification, dividing cropland into dry land and paddy fields for separate analyses of snow cover impacts.

Northeast China lost 6,694 km$^2$ of croplands between 2000 and 2020, primarily due to urban expansion and the Grain for Green Program. To minimize potential confounding effects from land use changes during the 2001 to 2020 period, our analysis focused on croplands with consistent land cover throughout the study period. Pixel-wise screening of land use distribution data from 2001 to 2020 extracted the croplands that remained unchanged throughout the 20 years for analysis. This approach

ensures that observed GPP variations can be more reliably attributed to snow cover dynamics rather than land use changes, thereby enhancing the robustness of the findings on snow cover-cropland interactions.

### 2.3 Methods

#### 2.3.1 Trend analysis

The long-term snow cover-GPP relationship in Northeast China was investigated by first calculating the annual averages of snow cover indicators (SWE, SCED, and SCD) and GPP from 2001 to 2020. Subsequently, we combined the Theil-Sen slope method (Sen, 1968) with the Mann-Kendall test (Kendall, 1948; Mann, 1945) to estimate trends in the time series of SCD, SWE, and GPP. The Mann-Kendall test is widely used to analyze trends in time series data as it addresses the serial

correlation issue and does not require a specific data distribution (Abebe et al., 2022; Xu et al., 2017). This method provided the trend magnitude and direction while helping to identify regions with significant changes between GPP and snow cover.

#### 2.3.2 Partial correlation

Partial correlation analysis statistically calculates the relationship between two variables while

controlling for the effects of one or more covariates (Gonzalez, 2003). It has recently been widely applied to study ecosystem response mechanisms and is particularly well-suited for disentangling causal relationships among complex environmental variables (Kashyap and Kuttippurath, 2024; Wei et al., 2022). Specifically, we applied pixel-by-pixel partial correlation to examine the impacts of SCD,





SWE, and SCED on GPP across different land use types. Meanwhile, this analysis eliminated the

effects of concurrent temperature, precipitation, and solar shortwave radiation to ensure that the

correlations between snow cover indicators and GPP reflected the direct impacts.

### 2.3.3 Ridge regression

This study identified the potential multicollinearity and interaction effects among snow cover indicators.

Multicollinearity can lead to unstable regression coefficients in ordinary least squares (OLS) regression,

potentially compromising the reliability of parameter estimates (Midi et al., 2010). This issue was

addressed with ridge regression, which introduced an L2 regularization into OLS regression (Zhao et

al., 2023). This approach effectively reduced variance in parameter estimates while introducing

minimal bias, making it particularly suitable for analyzing correlated environmental variables.

We employed ridge regression to quantify the relative contributions of snow cover indicators across

different zones through pixel-wise calculations, enabling spatially explicit analysis of snow cover-GPP

relationships. This method identified the dominant snow cover indicators influencing cropland GPP in

each zone, providing valuable insights into the spatial heterogeneity of snow cover's effects on

vegetation productivity. Ridge regression was implemented with careful consideration of regularization

parameters to ensure optimal model performance across various zones.

### 2.3.4 Savitzky-Golay filter

Savitzky-Golay (SG) filter is a smoothing technique based on local polynomial fitting, which smoothes

signals while preserving high-order derivative information. The SG filter determines the smoothed

result within a local window by least squares fitting of the data points in it. It is particularly suitable for

processing data with significant noise, as it effectively reduces noise while maintaining the

fundamental characteristics of the signal(Ruffin and King, 1999). The advantage of the SG filter lies in

its ability to smooth the data while preserving high-frequency information of the signal. such as the

slope and curvature. Consequently, it is widely used in signal processing, spectral analysis, and

chemical data analysis (Sa-ing et al., 2018; Sadıkoglu and Kavalcıoğlu, 2016).



Based on all valid cropland pixels in Northeast China, a scatter plot illustrating the relationship between snow cover indicators and GPP was constructed. The 95th percentile of GPP was extracted within each snow cover indicator interval to reduce the influence of extreme values. An SG filter was then applied to smooth the 95th-percentile response curve. Finally, the snow cover indicator corresponding to the peak of the smoothed curve was defined as the optimal snow cover threshold for GPP.

**2.3.5 Partial least squares structural equation model**

The direct and indirect mechanisms by which snow cover affects GPP were thoroughly investigated by constructing a partial least squares structural equation model (PLS-SEM) to estimate complex causal relationships among multiple variables. This powerful SEM explores the interactions between observed and latent variables and is suitable for non-normally distributed and small-sample data (Hair et al., 2019; Luo et al., 2017). Two latent variables were created during model construction: "Snow" and "Climate." "Snow" included three snow cover indicators: SCD, SWE, and SCED, whereas "Climate" included three meteorological indicators: precipitation, temperature, and solar radiation. The PLS-SEM also included two soil parameter variables (ST and SM) and GPP data for different vegetation types. PLS-SEM was applied to all six different regions in Northeast China. All variable data were normalized before path analysis to eliminate scale differences among variables and enhance model stability. Ultimately, three paths were constructed to reflect the multiple impacts of snow cover on the GPP of different vegetation types across the six regions in Northeast China.

The path coefficients in PLS-SEM analysis represent the magnitude and direction of direct effects between two variables. Positive and negative path coefficients correspond to the positive and negative impacts of the independent variable on the dependent variable, respectively, with their values quantifying the impact strength. The goodness-of-fit (GOF) index globally evaluates the quality of the path models and determines their validity. A GOF above 0.36 indicates applicable model results (Wetzels et al., 2009).



## 3 Results

### 3.1 The spatiotemporal dynamics of snow cover

Figure 2 displays the spatial distribution of mean SWE in Northeast China from HY2000 to HY2020. Among the six sub-regions, the mean SWE of Sanjiang Plain was the highest at 9.66 mm. The high SWE values were observed in the Khingan Range-Nenjiang conjunction and the Sanjiang Plain. The Khingan Range had the second-highest mean value of 9.05 mm and the second-highest maximum value of 20.10, followed by Songnen Plain. The Changbai Mountain also had a relatively high mean SWE of 6.26 mm. The semi-arid Western Sand Area had the second-lowest mean SWE but the highest maximum SWE. The mountainous areas (Changbai Mountain and Khingan Range) had greater snowfall than the other five sub-regions, explaining the high SWE levels. The Liaohe Plain had the lowest SWE among these six sub-regions due to its lower latitude. The negative slope of the SWE fitting line in Figure 2(c) indicates a slight decreasing trend from HY2001 to HY2020. As shown in Figure 2(d), the area exhibits a decreasing trend that accounts for about 65% of the total area, with dry land covering 60.21% and paddy fields covering 5.01%.

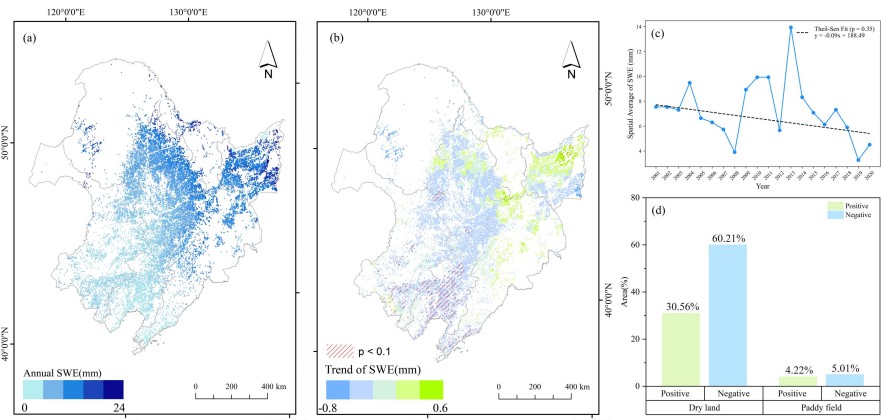

Figure 2 The spatial and temporal changes of SWE in Northeast China from HY2000 to HY2020: (a) spatial distribution of mean SWE; (b) changing trend of SWE, the green areas represent positive impacts, while the blue areas indicate negative impacts; and the shaded regions denote pixels that were significant at the p < 0.1 level; (c) annual changes of SWE; (d) the comparison of dry land and paddy land.



Figure 3 shows the spatial distribution of mean SCD in Northeast China from HY2000 to HY2020. The SCD of the whole region ranged from 0 to 159.47 days, averaging 79.99 days, 58% of which exceeded 80 days. High SCD values were primarily observed in the northeastern and mountainous areas, while low SCD values were distributed in the southwest and low-altitude areas. Overall, a decreasing trend was observed from the northeast to the southwest. Noticeable differences were observed in Changbai Mountain. Its northeastern area was closer to the Sanjiang Plain and showed significantly higher SCD than the southwestern area near the Liaohe Plain. The Khingan Range had the highest mean SCD of 120.29 days and the highest maximum SCD of 159.47 days, higher than the Changbai Mountain. The Sanjiang Plain had the second-highest mean SCD of 110.85 days, followed by the Songnen Plain. The negative slope of the fitting line in Figure 3(c) indicates a decreasing trend in SCD, with significant fluctuations between 2008 and 2014. Meanwhile, 54.1% of the croplands in the northeastern area experienced extended SCD, which included 49.02% of dry land and 5.16% of paddy fields, primarily distributed in the Songnen Plain. Areas with shortened SCD accounted for 45.9% of the total area, including 40.89% of dry land and 4.93% of paddy fields. Areas with significant SCD declines were mainly concentrated in the Liaohe Plain, similar to the spatial distribution of interannual SWE variation trends.

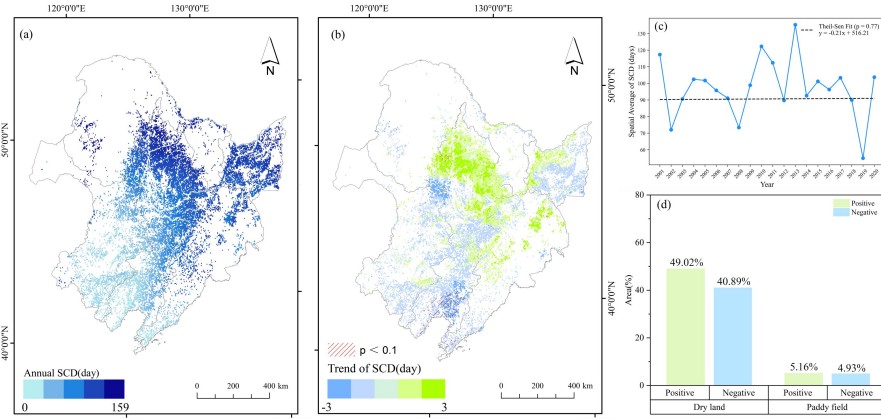

Figure 3 The spatial and temporal changes of SCD in Northeast China from HY2000 to HY2020: (a) spatial distribution of mean SCD; (b) changing trend of SCD, the green areas represent positive impacts, while the blue areas indicate negative impacts; and the shaded regions denote pixels that were significant at the p < 0.1 level; (c) annual changes of SCD; (d) the comparison of dry land and paddy land.



Figure 4 displays the spatial distribution of mean SCED in Northeast China from HY2000 to HY2020. The SCED distribution pattern was similar to that of SWE and SCD. Higher SCEDs were still primarily observed in the northeastern and mountainous areas, while lower SCEDs were distributed in the southwest and low-altitude areas. Statistical results showed that SCED ranged from 0 to 222.74

280    days, averaging approximately 164.19 days. About 49% of the pixels had SCD ranging from 180 to 210 days, while 49% of the pixels had SCEDs extending into March of the following year. Such pixels were concentrated in the Changbai Mountain, Sanjiang Plain, and the northeastern Songnen Plain. Pixels with SCEDs above 210 days accounted for only about 3% and were mainly distributed in the Khingan Range. The average SCED in the Sanjiang Plain and Khingan Range were approximately 195.08 days

and 196.85 days, respectively. According to Figure 3-6(b), the slope of the SCED fitting line in Figure 3(c) indicates a slight advancing trend. The SCED trend was relatively stable between 2005 and 2007, while fluctuations ranging from 10 to 30 days were observed in other years. Delayed SCEDs were observed in 61% of the areas, with 54.02% being dry land and 5.78% being paddy fields. Such areas were primarily distributed in the Songnen Plain, Sanjiang Plain, and Changbai Mountain. Only 39% of

the areas exhibited earlier SCEDs, with 34.47% being dry land and 5.73% being paddy fields, mainly concentrated in the Liaohe Plain.

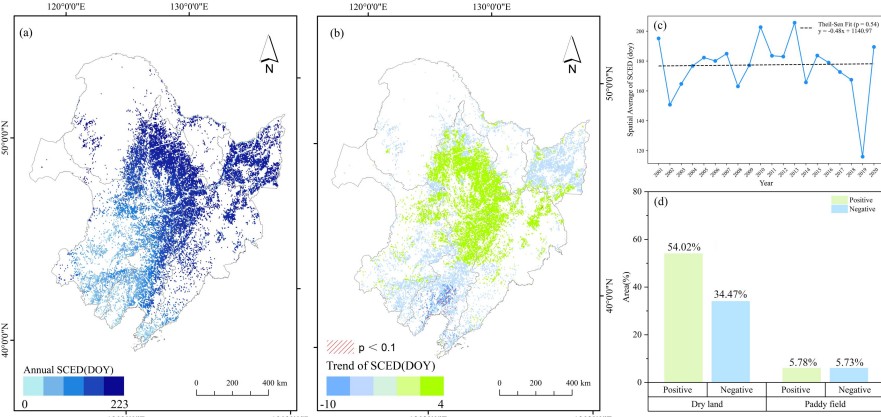

Figure 4 The spatial and temporal changes of SCED in Northeast China from HY2000 to HY2020: (a) spatial distribution of mean SCED; (b) changing trend of SCED, the green areas represent positive

impacts, while the blue areas indicate negative impacts; and the shaded regions denote pixels that were significant at the p < 0.1 level; (c) annual changes of SCED; (d) the comparison of dry land and paddy



land.

These spatiotemporal patterns of snow cover indicators provide crucial insights into the regional variability of snow cover dynamics and their potential impacts on vegetation productivity in Northeast China. The observed trends reflect the complex interactions between climate change and local geographical factors, thus requiring region-specific adaptation strategies in agricultural management and ecosystem conservation practices.

Table 2 Statistics of SWE, SCD and SCED in six geographical regions in Northeast China

| Geographic region | SWE (mm) | | | SCD (day) | | | SCED (day) | | |
|---|---|---|---|---|---|---|---|---|---|
| | Max | Mean | Mini | Max | Mean | Mini | Max | Mean | Mini |
| SN | 13.51 | 5.64 | 2.19 | 152.95 | 94.99 | 25.97 | 215.30 | 180.59 | 23.63 |
| SJ | 19.40 | 9.66 | 2.72 | 135.64 | **110.85** | 17.15 | 211.86 | **195.08** | 20.72 |
| LH | 5.04 | 1.73 | 1.01 | 98.29 | 36.25 | 20.87 | 187.09 | 125.81 | 34.84 |
| CB | 13.97 | **6.26** | 2.66 | 135.48 | 87.69 | 31.25 | 212.55 | 176.09 | 38.27 |
| WW | 24.07 | 2.35 | 1.77 | 156.25 | 32.03 | 22.02 | 217.99 | 108.54 | 31.31 |
| XAL | **20.10** | 9.05 | 3.41 | **159.47** | **120.29** | 24.75 | **222.74** | **196.85** | 21.84 |

**3.2 The relationship between snow cover and soil properties**

Figure 5 illustrates the spatial distribution of correlation coefficients between winter SWE and soil parameters (ST and SM) of the subsequent year. As shown in Figure 5(a), the correlation coefficient between SWE and ST ranges from –0.78 to 0.70, indicating predominantly negative influences of SWE on the ST in the subsequent year. Areas with negative correlations accounted for approximately 85% of the total croplands, and areas with significantly negative correlations concentrated in the Songnen Plain and Sanjiang Plain. A small portion of areas with positive correlation was found in the Western Sand Area and northern of Changbai Mountain. According to Figure 5(b), the correlation coefficient between SWE and SM ranges from –0.59 to 0.88, indicating a primarily positive impact of SWE on the SM in the subsequent year. Areas with positive correlations accounted for about 61% of the total croplands, whereas areas with significantly positive correlations mainly concentrated in the Sanjiang Plain and Changbai Mountain. Areas with negative correlations between SWE and SM accounted for




approximately 39%, mainly concentrated in the Songnen Plain and Liaohe Plain.

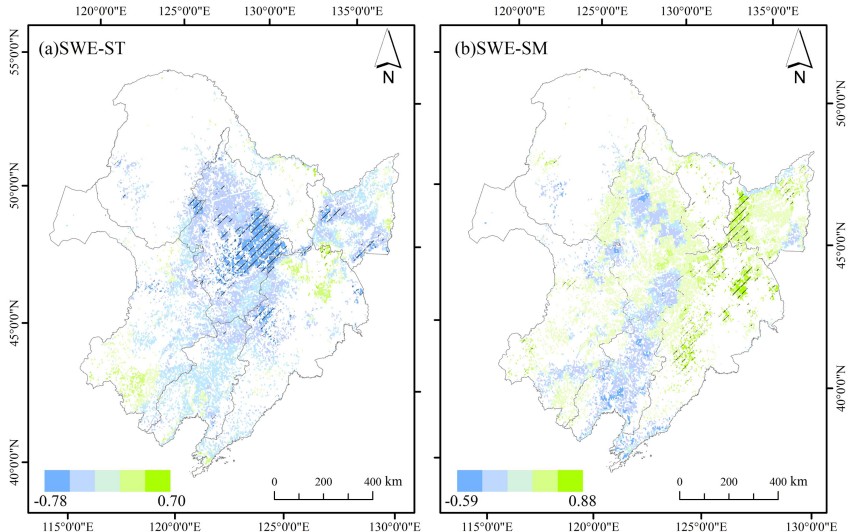

Figure 5 The correlation coefficients between SWE and soil properties: (a) soil temperature; (b) soil moisture. Blue and green pixels represent negative and positive correlations respectively. The shaded regions denote pixels that were significant at the p < 0.1 level.

Figure 6 illustrates the spatial distribution of correlation coefficients between winter SCD and soil parameters (ST and SM) of the subsequent year. As shown in Figure 6(a), the correlation coefficient between SCD and ST ranges from –0.85 to 0.57, indicating that the influence of SCD on the subsequent year's ST is predominantly negative. Areas with negative correlations accounted for approximately 90% of the total croplands, while areas with significantly negative correlations primarily concentrated in the Songnen Plain and Sanjiang Plain. These results suggest that a longer SCD results in slower soil warming in the following spring. Only a small number of areas with positive correlations were found in the northern part of Changbai Mountain and the Western Sand Area. Thus, a longer SCD in these areas may, to some extent, promote ST recovery through insulating effects. According to Figure 6(b), the correlation coefficient between SCD and SM ranges from –0.56 to 0.75, indicating a primarily positive impact of SCD on the subsequent year's SM. Areas with positive correlations accounted for 78%, and areas with significantly positive correlations mainly concentrated in the Songnen Plain and Changbai Mountain. Areas with negative correlations accounted for only 22%, and



SCD's moderating effect on SM transitions from positive to negative from northeast to southwest. The

pixels with SCD negatively affecting SM were primarily concentrated in Liaohe Plain.

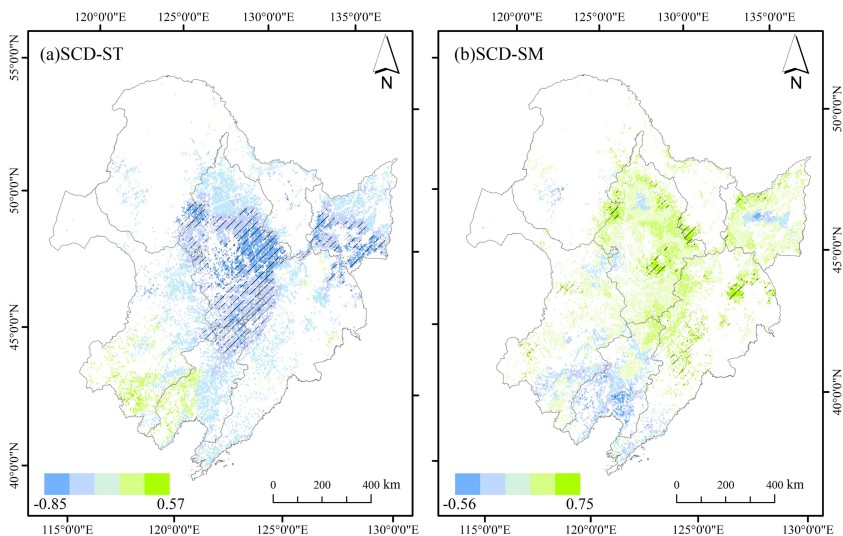

Figure 6 The correlation coefficients between SCD and soil properties: (a) soil temperature; (b) soil

moisture. Blue and green pixels represent negative and positive correlations respectively. The shaded

regions denote pixels that were significant at the p < 0.1 level.

Figure 7 illustrates the spatial distribution of correlation coefficients between winter SCED and soil

parameters (ST and SM) of the subsequent year. Figure 7(a) shows that the correlation coefficient

between SCED and SM ranges from –0.74 to 0.56, indicating a primarily negative impact of SCED on

ST. Areas with negative correlations accounted for approximately 87% of the total croplands, and areas

with significant negative correlations mainly concentrated in the Songnen Plain and Sanjiang Plain. A

small number of areas with positive correlations were primarily found in the northern of Changbai

Mountain and the Western Sand Area. According to Figure 7(b), the correlation coefficient between

SCED and SM ranges from –0.62 to 0.68, indicating a mainly positive impact of SCED on the

subsequent year's SM. Areas with positive correlations accounted for approximately 61% of the total

croplands, while areas with negative correlations accounted for about 39%. The pixels with negative

SCED effects on SM were primarily concentrated in the Sanjiang Plain, Songnen Plain, and Liaohe

Plain.

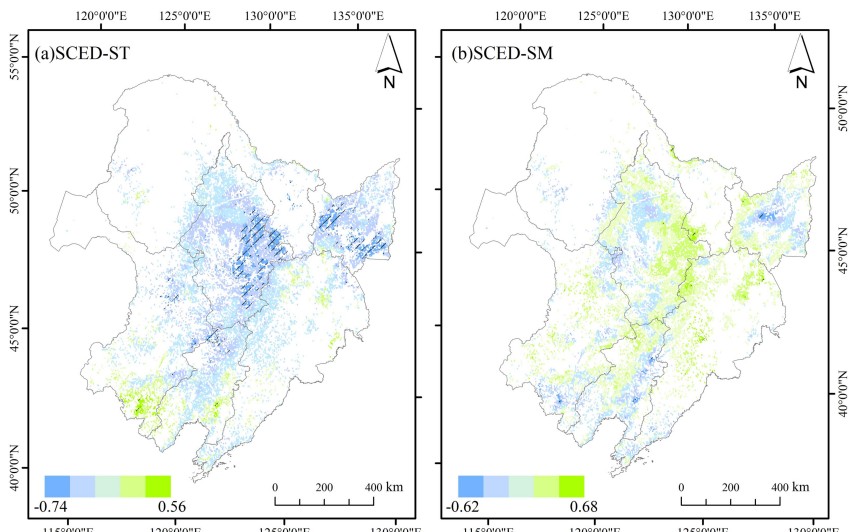

Figure 7 Spatial pattern distribution results of the relationship between SCED and soil properties: (a) soil temperature; (b) soil moisture. Blue and green pixels represent negative and positive correlations respectively. The shaded regions denote pixels that were significant at the p < 0.1 level

### 3.3 The relationship between GPP and snow cover

Figure 8 displays the spatial distribution of GPP in Northeast China from HY2000 to HY2020, and Table 3 lists the statistical results of GPP in the six sub-regions. Cropland GPP generally shows a relatively uniform distribution pattern, as shown in Figure 8(a). The Changbai Mountain, Sanjiang Plain, and Khingan Range had relatively high GPP, followed by Songnen Plain and Liaohe Plain, while the Western Sand Area had the lowest GPP during the past 20 years. The interannual variation trends in Figure 8(b) indicate that over the past 20 years, 98% of the cropland shows an increasing trend in GPP, with significant GPP growth in 74% of the areas. Furthermore, the growth rates varied across different regions, with the GPP in the Western Sand Area increasing the fastest at an average of approximately 9.04 g·C/m² per year. According to Figure 8(d), 98% of the croplands show an increasing trend in GPP, with 89.1% being dry land and 9.44% being paddy fields. Moreover, 74% of the areas showed significant GPP growth. Only 2% of the areas exhibited a declining trend in GPP, with 1.16% being dry land and 0.30% being paddy fields. Such areas were primarily concentrated in the southern part of the Liaohe Plain and the eastern part of the Sanjiang Plain.



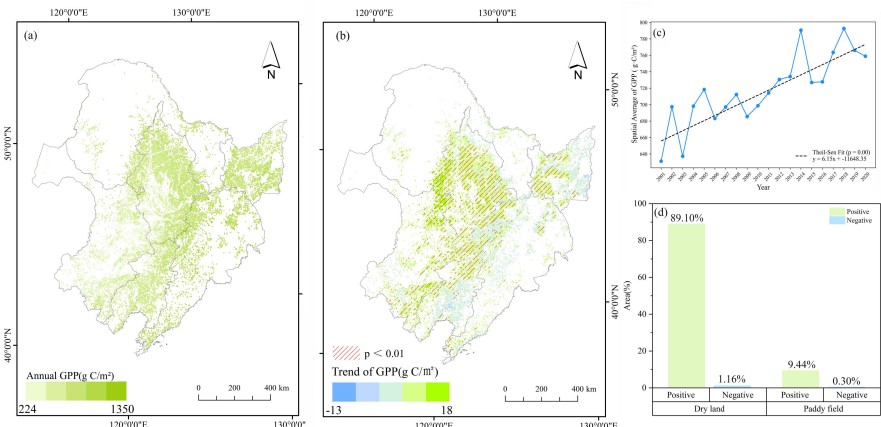

Figure 8 The spatial and temporal changes of GPP in Northeast China from HY2000 to HY2020: (a) spatial distribution of mean GPP; (b) changing trend of GPP the green areas represent positive impacts, while the blue areas indicate negative impacts; and the shaded regions denote pixels that were significant at the p < 0.1 level; (c) annual changes of GPP; (d) the comparison of dry land and paddy land.

Table 3 GPP statistics of six geographic regions in Northeast China

| Geographic regions | Max. (g·C/m²) | Mean. (g·C/m²) | SD (g·C/m²) |
|---|---|---|---|
| SN | 974.99 | 606.33 | 78.50 |
| SJ | 1111.97 | 701.92 | 82.34 |
| LH | 1034.13 | 612.78 | 76.29 |
| CB | 1350.56 | 799.75 | 125.95 |
| WS | 828.16 | 481.57 | 86.89 |
| XAL | 1048.16 | 690.71 | 108.01 |

Figure 9 displays the partial correlation between SWE and GPP. Approximately 56.79% of the dry land and 6.27% of the paddy fields showed positive correlations between GPP and SWE, with significant correlations in 8.3% of the dry land and 10.0% of the paddy fields. Areas with positive correlations were concentrated in the northern part of the Sanjiang Plain and the central-western part of the Songnen Plain. In contrast, areas with negative correlations were mainly in the southern part of the Liaohe Plain, accounting for 6.2% of the total dry lands. The effects of SWE on GPP are complex. For one thing, snowmelt enhances SM and promotes early crop growth with adequate water (Li et al., 2025;





Pan et al., 2022). For another, excessive SWE leads to saturated SM after snowmelt and denies oxygen

to the roots, thereby affecting plant growth and health (Liu et al., 2023; Wang et al., 2024).

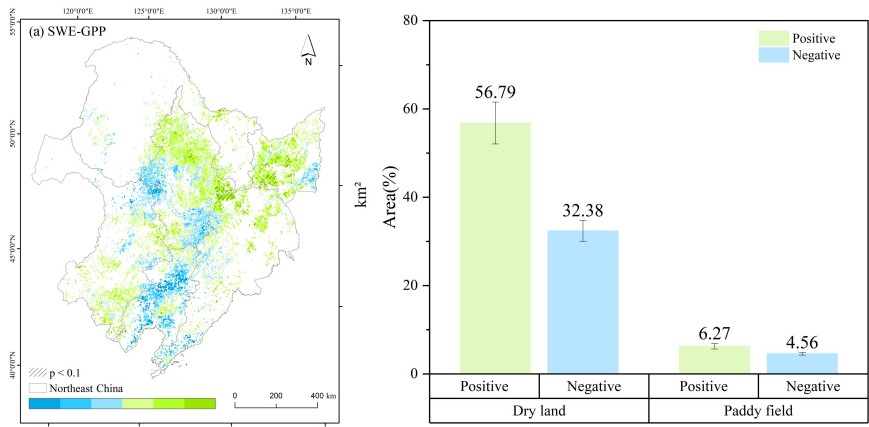

Figure 9 Spatial distribution and area statistics of the relationship between SWE and cropland GPP

from HY2001 to HY2020: (a) spatial distribution of partial correlation coefficients, where blue and

green represent pixels with negative and positive correlations, respectively; (b) area percentage

statistics, with the range of error bars indicating the area with significant impact ($p < 0.1$).

Figure 10 shows the partial correlation between SCD and GPP. SCD is the main snow cover indicator

affecting GPP in dry lands. The dry lands with positive correlations between SCD and GPP accounted

for 66.27% of the total area, while the paddy fields with positive correlations accounted for 6.79% of

the total area. Areas with significant responses were mainly concentrated in the Songnen Plain,

accounting for 11.0% of the total dry lands. The positive effect of SCD on cropland GPP exhibited a

latitudinal gradient, with gradually increasing response intensities as latitude increases. Long-term

snow accumulation effectively protects plants and soil from extreme weather conditions, providing a

relatively warm and stable growth environment. Noteworthy, the paddy fields exhibited relatively low

sensitivity to SCD, accounting for only 8.0% of the pixels with significant positive correlations,

indicating a limited isolation effect of snow on paddy fields. Paddy field productivity relies more on

human irrigation management and other climatic factors.



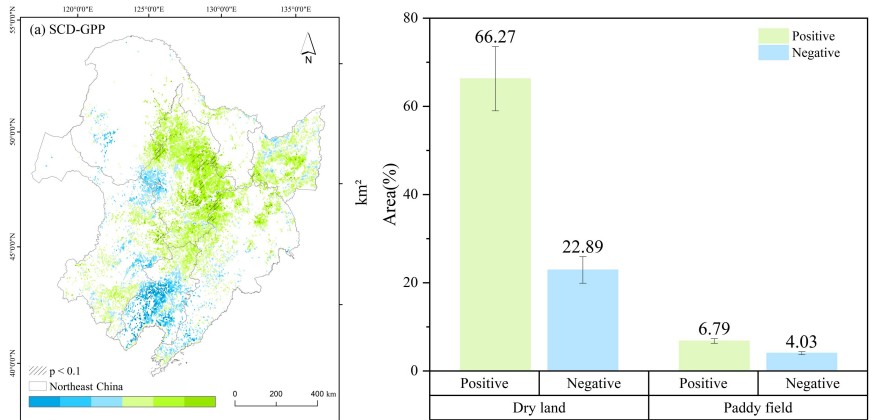

Figure 10 Spatial distribution and area statistics of the relationship between SCD and cropland GPP
from HY2001 to HY2020 (a)spatial distribution of partial correlation coefficients, where blue and
green represent pixels with negative and positive correlations, respectively; (b) area percentage
statistics, with the range of error bars indicating the area with significant impact (p < 0.1).

Figure 11 displays the partial correlation between SCED and GPP. The regulatory effect of SCED on
GPP is more pronounced in paddy fields. Specifically, 14.0% of the paddy fields exhibited significant
positive correlations between SCED and GPP, notably higher than the 7.3% observed in dry lands.
Further analysis reveals that areas with SCED negatively correlated with dry land GPP constituted
31.26% of the total area, while the corresponding proportion for paddy fields is 6.51%. Areas
exhibiting negative correlations were primarily concentrated in the Liaohe Plain and the eastern part of
the Sanjiang Plain. Early SCEDs may lead to unfavorable climatic conditions during the early growth
stage of plants, potentially impacting their growth and survival rates. Conversely, delayed SCED can
mitigate the loss of spring runoff due to snowmelt and better align crop transplantation with the optimal
ST window for crop growth.



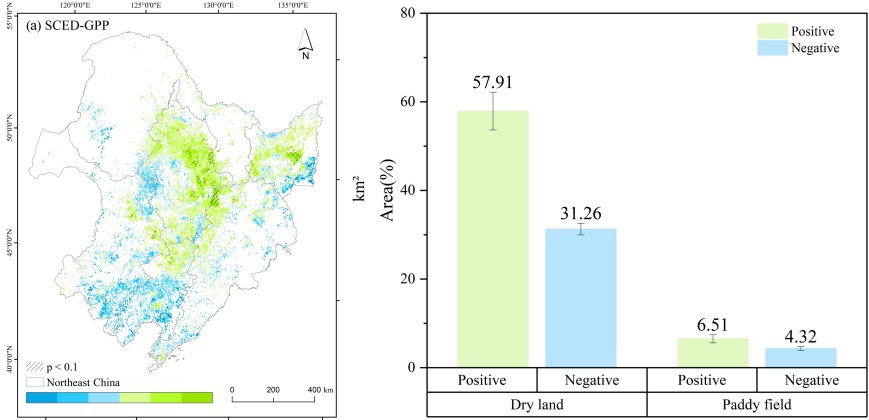

Figure 11 Spatial distribution and area statistics of the relationship between SCED and cropland GPP from HY2001 to HY2020: (a) spatial distribution of partial correlation coefficients, where blue and green represent pixels with negative and positive correlations, respectively; (b) area percentage statistics, with the range of error bars indicating the area with significant impact (p < 0.1).

## 4 Discussion

### 4.1 Dominant snow cover indicators in the snow cover-GPP correlation

Figure 12 presents the spatial distribution and area proportions of the relative contributions of different snow cover indicators to cropland GPP in Northeast China. SWE predominantly drove cropland GPP variations in the Western Sand Area and Liaohe Plain, which accounted for approximately 50% of the GPP changes, significantly higher than the contributions of SCD and SCED. In contrast, SCED emerged as the primary driver in the Changbai Mountain, Sanjiang Plain, and Khingan Ranges, with contribution rates reaching 45.2%, 49.5%, and 38.6%, respectively. The Songnen Plain demonstrated a distinct pattern, with SCD dominating within 39.59% of the total area, substantially higher than SWE (31.29%) and SCED (29.11%). This regional analysis elucidated spatial heterogeneity in the relative contributions of snow cover indicators to cropland GPP variations across Northeast China. The findings demonstrated distinct geographical zoning characteristics that provided a theoretical foundation for understanding the differential impacts of snow cover changes on agricultural productivity across regions. SWE exerted more significant influences in relatively arid areas, while the impacts of SCED



manifested more prominently in colder areas.

Notably, SWE dominates GPP variability in moisture-limited areas, like the Western Sand Area and the
Liaohe Plain, accounting for ~50% of the observed fluctuations. Its contribution was 1.6- to 1.7-fold
greater than those of SCD and SCED. These results aligned with hydrological theory positing that SWE
is a critical drought-mitigating reservoir in arid ecosystems through delayed meltwater release (Barnett
et al., 2005). Conversely, SCED emerged as the principal driver in colder high-latitude areas (Changbai
Mountain, Sanjiang Plain, Khingan Range), explaining 38.6% to 49.5% of GPP variations. Such spatial
patterns likely reflect SCED's bidirectional effects in regulating growing season onset via albedo
modulation and frost protection through insulation effects (Pulliainen et al., 2020). Sanjiang Plain
exhibited hybrid behavior, where the SCD predominance (39.59%) suggested intermediate sensitivity
to SCD and hydrologic inputs.

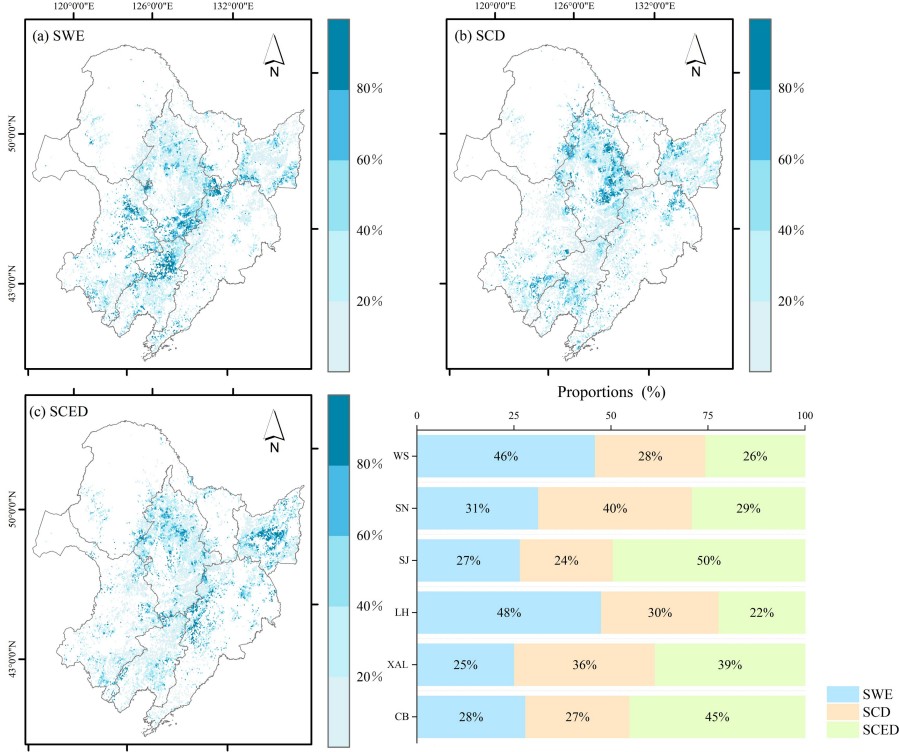

Figure 12 Spatial distribution and area percentage of snow-related indicators driving cropland GPP
variation in different regions of Northeast China from 2001 to 2020.




### 4.2 The thresholds of snow cover-GPP correlation

As depicted in Figure 13(a), GPP demonstrates phased responses to the increasing SWE. The initial SWE accumulation (0 to 10.36 mm) induced substantial GPP enhancement through improved SM availability, and the photosynthetic activity peaked at the SWE of 10.36 mm. Beyond this hydrological

optimum, the SWE-GPP correlation degraded into a stable plateau phase (SWE = 15 to 25 mm), where additional snow cover contributed minimally to productivity gains. Figure 13(b) reveals a parabolic relationship between SCD and cropland GPP. Short SCD regimes (< 20 days) induced GPP suppression due to inadequate frost protection and SM deficits. As SCD extended to the bioclimatic optimum (132.79 days), GPP recovered gradually via extended vernalization and reduced freeze-thaw cycles.

However, excessive SCD (> 133 days) triggered a 0.7% daily GPP decline through photoperiod limitations and delayed phenological development. SCED exhibits threshold-mediated impacts, as shown in Figure 13(c). Early SCEDs (DOY < 100) corresponded to 25% to 30% GPP reduction from premature exposure to spring frost events. The critical transition at DOY 207.13 marked the onset of positive SCED-GPP coupling, attributable to improved alignment between snowmelt timing and

vegetation green-up requirements. Post-DOY 207 stabilization reflected optimized thermal-moisture conditions sustaining peak photosynthetic rates.

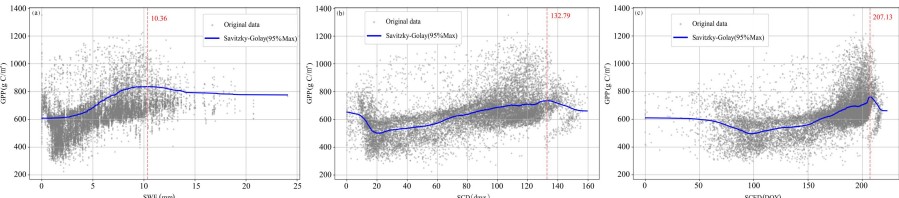

Figure 13 The critical threshold for the influence of different snow-related indicators on cropland GPP from HY2001 to 2020:(a) SWE; (b) SCD; (c)SCED. Gray represents the scatter plot of multi-year

averages of snow-related indicators and GPP; blue represents the fitting curve; the red vertical dashed line indicates the optimal threshold.

### 4.3 The mechanisms underlying snow cover impacts on GPP

Figure 14 systematically quantifies the influence of snow cover on GPP, and the snow cover-GPP causal networks exhibit profound spatial heterogeneity across the six agroecological regions with dry





land and paddy systems. Regarding dry land systems, Sanjiang Plain exhibited the strongest snow cover-GPP facilitation ($\beta$ = 0.47), followed by the Khingan Range ($\beta$ = 0.29) and Songnen Plain ($\beta$ = 0.26). In contrast, significant suppression was observed in Liaohe Plain ($\beta$ = –0.23), Western Sand Area ($\beta$ = –0.16), and Changbai Mountain ($\beta$ = –0.10), attributable to snowmelt-induced nutrient leaching. Universal snow cover-ST negative coupling (mean $\beta$ = –0.58 ± 0.12) reduced growing-season thermal

advantages. In the Changbai Mountain agroecosystem, the snow cover-ST-GPP pathway exhibited a significant indirect effect ($\beta_{snow-ST} \times \beta_{ST-GPP}$ = –0.67 × 0.76 = –0.51), indicating that snow cover suppressed photosynthetic efficiency through thermal limitation mechanisms during the growing season. The snow cover-SM interaction exhibited marked spatial heterogeneity across the regions due to divergent hydrological mechanisms. In the Liaohe Plain dry land systems, snow cover demonstrated

a negative coupling with SM ($\beta$ = –0.16), subsequently amplifying GPP suppression through SM's strong positive linkage to productivity ($\beta$ = 0.96). Similar snow-SM-GPP inhibition patterns were observed in the Songnen Plain ($\beta_{snow-SM}$ = –0.11).

In terms of paddy ecosystems, the Sanjiang Plain maintained the strongest direct snow cover-GPP facilitation ($\beta$ = 0.57), followed by the Khingan Range ($\beta$ = 0.24) and Songnen Plain ($\beta$ = 0.02).

Conversely, snow cover negatively correlated with GPP in the Liaohe Plain ($\beta$ = –0.28), Changbai Mountain ($\beta$ = –0.06), and Western Sand Area ($\beta$ = –0.18). ST mediated contrasting effects. Positive ST-GPP coupling dominated in the Songnen Plain ($\beta$ = 0.83), Sanjiang Plain ($\beta$ = 0.70), Khingan Range ($\beta$ = 0.63), and Changbai Mountain ($\beta$ = 0.58). Negative ST-GPP relationships emerged in the Liaohe Plain ($\beta$ = –0.79) and Western Sand Area ($\beta$ = –0.73), where elevated ST intensified evapotranspiration

losses (ET/PET ratio > 1.2), exacerbating soil desiccation and photosynthetic constraints (Bodner et al., 2015; Zambreski et al., 2018). This aridity amplification mechanism explained 68% of GPP variance in water-limited paddies. Within paddy systems, SM maintained consistent positive correlations with GPP (mean $\beta$ = 0.91 ± 0.05). However, in the Songnen Plain and Liaohe Plain, the snow cover-SM-GPP pathway exhibited significant indirect suppression effects, quantified as –0.097 ($\beta_{snow-SM}$ = –0.11 ×

$\beta_{SM-GPP}$ = 0.88) and –0.281 ($\beta_{snow-SM}$ = –0.30 × $\beta_{SM-GPP}$ = 0.94), respectively. These negative mediating effects indicate that snow cover reduces irrigation water availability through competitive consumption of meltwater resources, thereby constraining rice photosynthetic capacity in these water-stressed agroecosystems.



Therefore, the impacts of snow cover on GPP exhibit pronounced spatial heterogeneity due to geographic-climatic conditions and cultivation regimes. These effects manifest through direct (positive/negative) forcing and indirect hydrothermal mediation pathways, forming a threshold-modulated regulatory framework where snow cover-derived hydrological subsidies and thermal constraints interactively shape photosynthetic efficiency gradients across agroecological zones.

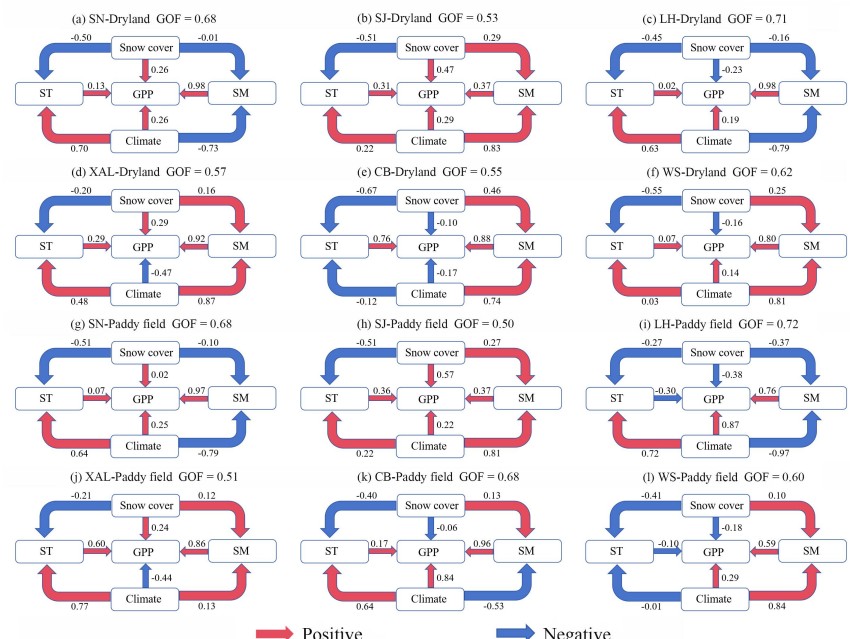

Figure 14 The standardized path coefficients between snow cover and GPP via soil properties. The Model fit was validated through goodness-of-fit (GOF), demonstrating acceptable parameter estimation accuracy. The red and bule arrows represents the negative and positive influence. Arrowhead orientation specifies causal pathways from exogenous to endogenous variables. Blue arrows denote inhibitory effects, whereas red pathways indicate facilitative relationships. Arrowhead orientation specifies causal pathways from exogenous to endogenous variables.

**5 Conclusion**

Utilizing multi-source remote sensing data from 2001 to 2020, this study systematically investigated the spatiotemporal patterns of snow cover dynamics (SWE, SCD, SCED) and GPP variations across



Northeast China, revealing marked regional heterogeneity in snow cover-crop interactions. The Liaohe

Plain and Western Sand Area exhibited significant SWE declines (–3.2 mm/decade, p<0.05), contrasting with agricultural heartlands like the Songnen Plain and Sanjiang Plain with increased SCD (+1.8 days/decade) and delayed SCED (+2.1 days/decade). Spatially, 65% of the croplands exhibited SWE reduction (mean –12.7 ± 3.1 mm), while 54.1% and 61% showed SCD prolongation (+7.3±2.4 days) and SCED retardation (+4.6±1.8 days), respectively. Concurrently, GPP demonstrated robust

upward trends, particularly in core production zones (Songnen Plain: +18.7%, Sanjiang Plain: +22.3%), where intensified SCD-SCED synergism enhanced growing-season hydrothermal optimization.

The interference from climatic factors, such as temperature, precipitation, and solar radiation, was excluded via partial correlation analysis to independently investigate the intrinsic snow cover-GPP relationships. Results indicated that SCD played a particularly significant role in regulating dry land

GPP, while SCED had a more prominent impact on paddy GPP. The relative contributions of snow cover indicators across different sub-regions were quantified through ridge regression for pixel-wise analysis of each metric's weight. The results revealed that SWE contributed most significantly to GPP variations in the Western Sand Area and Liaohe Plain, whereas SCD changes dominated GPP variations in the Songnen Plain and Sanjiang Plain. The Changbai Mountain and Khingan Range were primarily

influenced by SCED variations. Additionally, threshold analysis identified the optimal ranges of cropland GPP responses to snow cover indicators: peak positive GPP facilitation was observed at the SWE of approximately 10.36 mm, the SCD of 132.79 days, and the SCED of 207.13 days (after September 1st). Exceeding or falling below these thresholds weakened or even reversed the positive effects.

The influence pathways of snow cover, climate, and soil factors on cropland GPP were comprehensively analyzed by constructing a PLS-SEM model to quantify the direct effects of snow cover indicators on GPP and their indirect effects mediated by SM and ST. Results demonstrated that SWE, SCD, and SCED directly affected GPP and indirectly influenced cropland productivity through complex interactions with climatic factors, i.e., regulating soil hydrothermal conditions. The model

analysis further revealed regional variations in the pathways across geographical subregions. The PLS-SEM outcomes provided scientific evidence for elucidating the snow cover-soil-vegetation coupling mechanisms in cold-region agroecosystems of Northeast China while offering theoretical support for regional agricultural management and climate change adaptation strategies.

In conclusion, this study focused on the GPP responses of dry lands and paddy fields in Northeast China to snow cover changes across different geographical partitions. However, inevitable human activities may interfere with the cropland analysis. Moreover, limitations in remote sensing data could omit potential pathways, such as the impact of snow cover on soil nutrients and plant photosynthetic efficiency. Additionally, this study did not consider different crops in croplands. A more comprehensive

study in the future could investigate the impact of snow cover on cropland GPP in different geographical environments by establishing experimental zones to quantify human activities such as fertilization and irrigation. Meanwhile, field observations can be adopted to further understand the impact of snow cover on soil nutrients and fertilizer utilization efficiency. Additionally, more detailed analyses could be achieved through higher-precision crop classifications.


**Author Contributions**

QY and HL formulated the original ideas presented in this manuscript. LL developed the methodology, performed the formal analysis and validation with QY, and drafted the original manuscript. XH curated the data. MC and YP conducted the investigation. QY and JC revised the manuscrip, and QY acquired

the funding. All authors contributed to the article and approved the submitted version.

**Funding**

This research was jointly supported by National Key Research and Development Program of China (Grant No. 2024YFD1500602-4).

**Acknowledgments**

This research is an output of Cropland Degradation Monitoring.

**Conflicts of Interest**

The authors declare no conflict of interest.



**Data Availability**

The data is available on request.

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
