# Peer review of "The influence of snow cover on gross"

_EGUsphere, 2025_

## Author Comment (AC1)

To editor:

We sincerely thank the reviewer for their valuable feedback and insightful comments. These have significantly improved the clarity, quality, and precision of our manuscript.

In response to the reviewers' comments, we have revised the manuscript and prepared a point-by-point response. For easy reference, the original comments are reproduced in blue, followed by our responses in black. All changes made to the manuscript text are highlighted in red.

The manuscript presented a detailed study on the impact of snow on the cropland gross primary productivity in northeast China, by using dataset retrieved from satellite remote sensing and reanalyzes and others. The topic is interesting, as the structure is clear together with the logic. However, a main problem of the current version is the absence of physical mechanisms because most explanations are based on statistical analysis. In addition, I have some specific comments as below, and hope the authors can still improve the manuscript accordingly.

**Response:** We deeply appreciate the reviewer's constructive and thoughtful feedback. The points raised have been instrumental in improving the manuscript, particularly in clarifying the physical mechanisms behind the observed relationships and in refining the structure and presentation. We made significant modifications, including: (1) removing the former results regarding paddy land, including method, figures, and context; (2) moving the former Discussion into Results and drafting a new Discussion. Below, we provide a detailed point-by-point response to each comment.

In the second paragraph of the introduction section, a summary on the previous studies on the impact of snow on GPP was presented. How about the associated study in the northeast China? I think the readers would wonder if there is any existed studies in the same area.

**Response:** Thank you for this helpful suggestion, and we add the comparison with previous work in the Introduction (Line 92-97) and Discussion (Line 504-509).

Previous work by Wang et al. (2024) shows that the relationship between snow cover and vegetation productivity in Northeast China varies by underlying surface type and is further modulated by local environmental conditions. A knowledge gap exists in previous work regarding how multi-metric snow characteristics interact with snow–vegetation productivity relationships simultaneously across agricultural regions in Northeast China. (Line 92-97)

Wang et al. (2024) showed that in this region, increases in SWE tend to favor GPP in dryland and grassland, while snow phenology metrics such as SCED and SCD are more influential in forests. Our results refine this picture by isolating cropland and demonstrating that (i) SWE dominates GPP variability in moisture-limited cultivated systems, (ii) SCD and SCED become critical where cold stress and drainage limitations are prominent, and (iii) the relative dominance of these metrics shifts systematically. (Line 504-509)

Reference:

Wang, Y. et al., 2024. Unraveling the effects of snow cover change on vegetation productivity: Insights from underlying surface types. Ecosphere, 15(5): e4855.

In figure 1, the six sub-regions in figure 1c is hard to clearly see. And in the context, only the climate features of northeast China was introduced, how about the six sub-regions? What do you mean "distinct climatic characteristics across these six sub-regions"? More explanations are necessary.

**Response:** We appreciate this comment and have revised both the figure and the text for clarity. In Figure 1c, we have enhanced the visual representation of the six sub-regions by adjusting the contrast and adding labels to improve readability. Additionally, we have expanded the text (Lines 121-129) to elaborate on the distinct climatic characteristics of these sub-regions, including differences in temperature, precipitation patterns, and snow cover duration, which influence GPP dynamics. The sub-regions were chosen based on these climatic differences to explore regional variations in snow cover's impact on GPP.

The region features a temperate monsoon climate characterized by winter snowfall (Xue et al., 2022), low evaporation rates, and high humidity. However, as shown in Table 1, there are pronounced climatic gradients across the six sub-regions. The effective accumulated temperature (≥10°C) ranges from 2320°C in the cooler Xing'an Mountain area to 3654°C in the warmer Liaohe Plain (Xu et al., 2023). Similarly, annual precipitation exhibits stark contrasts, from a mere 200–400 mm in the arid Western Sand Area to 800–1200 mm in the humid Changbai Mountain. These geographic and climatic differentiations are crucial for understanding regional ecosystem responses. (Lines 121-129)

[Figure]

**Figure 1 The overview of the study area: (a) China; (b) cultivated land in Northeast China; (c) digital elevation model (DEM) provided by SRTM; (d) land use types provided by LUCC**

In the method section, I think it may be not necessary to present all statistical algorithms because some of them are widely-employed ones. Besides, the spatial and temporal resolutions of these data are different, how to interpolate them into the same grids? And will that interpolation introduce any associated uncertainty into the analysis?

**Response:** We agree with the reviewer's comment regarding the level of detail in the methods section. We have streamlined the presentation of statistical algorithms to focus on those that are critical to the analysis, while removing more widely used ones.

Besides, we fully acknowledge the importance of data preprocessing methods. All datasets were resampled to a consistent spatial resolution of 0.05°×0.05° using the nearest neighbor method to facilitate subsequent pixel-by-pixel analysis. We recognize that resampling introduces uncertainties; however, it is a necessary step for multi-source data fusion and pixel-by-pixel statistical analysis, which are discussed in the Discussion 4.2 (Line 549-552).

**2.3.1 Trend analysis**

The long-term trends in the annual time series of SCD, SWE, and GPP (2001-2020) were analyzed using the Theil-Sen slope method (Sen,1968). The statistical significance of these trends was evaluated with the Mann-Kendall test (Kendall, 1948; Mann, 1945).

**2.3.2 Partial correlation**

All datasets were resampled to a consistent spatial resolution of 0.05°×0.05° using the nearest neighbor method to facilitate subsequent pixel-by-pixel analysis. Then, a partial correlation analysis was employed to statistically quantify the relationship between two variables while controlling for the effects of one or more covariates (Gonzalez, 2003; Kashyap and Kuttippurath, 2024; Wei et al., 2022). Specifically, we applied pixel-wise partial correlation to examine the impacts of SCD, SWE, and SCED on GPP across various land-use types, while controlling for concurrent temperature, precipitation, and solar radiation to isolate the direct effects of snow cover.

**2.3.3 Ridge regression**

Given the potential multicollinearity among snow cover indicators, we used ridge regression rather than ordinary least squares to ensure stable coefficient estimates (Zhao et al., 2023). This approach was applied pixel-wise to quantify the relative contributions of SCD, SWE, and SCED to GPP across the study area. This method identified the dominant snow-cover indicators influencing GPP on cultivated land in each zone, providing valuable insights into the spatial heterogeneity of snow cover's effects on vegetation productivity.

**2.3.4 Partial least squares structural equation model**

To decipher the complex causal pathways through which snow affects GPP, we employed a Partial Least Squares Structural Equation Model (PLS-SEM). Two latent variables were created during model construction, including Snow and Climate. Snow included three snow cover indicators covering SCD, SWE, and SCED, whereas Climate included three meteorological indicators covering precipitation, temperature, and solar radiation. The PLS-SEM also included two soil parameter variables, covering SM, ST, and GPP data for different vegetation types. All variables were normalized before the analysis to facilitate comparison of path coefficients. The path coefficients in PLS-SEM analysis represent the magnitude and direction of direct effects between two variables. Positive and negative path coefficients correspond to the positive and negative impacts of the independent variable on the dependent variable, respectively, with their values quantifying the impact strength. The goodness-of-fit (GOF) index globally evaluates the quality of the path models and determines their validity. A GOF above 0.36 indicates applicable model results (Wetzels et al., 2009).

From section 3.1, spatiotemporal distribution of snow cover is more suitable for the manuscript than "snow cover dynamics" in the title, because the latter is within a small scale as I see.
**Response:** Thank you for this suggestion. We have revised the title of Section 3.1 to "Spatiotemporal Distribution of Snow Cover" to reflect the scope of the analysis more accurately. The focus of this section is indeed on the distribution patterns of snow cover over time and space, rather than on its dynamics at a small scale (Lines 244).

I do not like the titles of section 3.2 and 3.3. Both are in the style of the relationship between A and B. However, the logical relation between A and B is not clear. I mean who is the reason and who is the result. For example, "the relationship between snow cover and soil properties" is somewhat amazing. Snow cover is associated with climate and weather, but soil property is by land surface. Does A decide B? or B decide A? If we only give some statistical results between A and B rather than the physical mechanism between them, this is only a mathematical game.
**Response:** We thank the reviewer for this insightful comment. We agree that the original section titles were too vague and did not reflect the causal relationships investigated. Following the reviewer's suggestion, we have revised the titles to represent the scientific logic of our analysis better:

Section 3.2 has been changed from "The relationship between snow cover and soil properties" to "Effects of Snow Cover on Soil Properties". (Line 307)

Section 3.3 has been changed from "The relationship between GPP and snow cover" to "Influence of Snow Cover on GPP". (Line 370)

I suggest the authors can add a short paragraph between section 4 and 4.1 to tell why and what will you do in the discussion section, and what is the logic among the three sections below.
**Response:** We have added a brief introductory paragraph between Section 4 and 4.1 to provide context for the discussion. This paragraph outlines the logical flow of the discussion, explaining how the results will be interpreted in light of the physical mechanisms at play and how each of the subsequent sections will contribute to understanding the impact of snow cover on GPP (Lines 479-488).

This study aimed to elucidate the spatiotemporal variations in snow cover parameters (SWE, SCD, and SCED) and their heterogeneous impacts on gross primary productivity (GPP) of cultivated land across six subregions in Northeast China, while uncovering the underlying regulatory mechanisms through soil properties. By integrating long-term satellite and reanalysis products with partial correlation analyses and a PLS-SEM framework, we show that snow cover exerts strong but spatially heterogeneous controls on cropland GPP via its effects on spring soil temperature and moisture, with distinct response patterns between dryland and paddy systems and among key agricultural plains. These findings demonstrate that snow is not merely a passive climatic background factor but an active regulator of agricultural carbon uptake in a region that is both snow-dominated and critical for national food security.

I suppose to see more physics in the section 4.3, but it is a pity that there are still some mathematical analyses rather than physical explanations. For example, "In the Changbai Mountain agroecosystem, the snow cover-ST-GPP pathway exhibited a significant indirect

effect ($\beta$snow-ST $\times$ $\beta$ST-GPP = –0.67 $\times$ 0.76 = –0.51), indicating that snow cover suppressed photosynthetic efficiency through thermal limitation mechanisms during the growing season." What I want to know is why and how the snow cover suppressed photosynthetic efficiency through some physical mechanisms.

**Response:** Thank you for raising this critical point. We have moved the former Discussion into Results, and added a new Discussion (Lines 478-579).

4 Discussion

This study aimed to elucidate the spatiotemporal variations in snow cover parameters (SWE, SCD, and SCED) and their heterogeneous impacts on gross primary productivity (GPP) of cultivated land across six subregions in Northeast China, while uncovering the underlying regulatory mechanisms through soil properties. By integrating long-term satellite and reanalysis products with partial correlation analyses and a PLS-SEM framework, we show that snow cover exerts strong but spatially heterogeneous controls on cropland GPP via its effects on spring soil temperature and moisture, with distinct response patterns between dryland and paddy systems and among key 
[revised manuscript text omitted]

---

## Author Comment (AC2)

**To Editor:**

We sincerely thank the reviewer for their valuable feedback and insightful comments. These have significantly improved the clarity, quality, and precision of our manuscript.

In response to the reviewers' comments, we have revised the manuscript and prepared a point-by-point response. For easy reference, the original comments are reproduced in blue, followed by our responses in black. All changes made to the manuscript text are highlighted in red.

This study analyzed the dynamic changes in snow cover and cropland GPP in Northeast China from 2001 to 2020. The need for such a study is well justified and the authors cite ample relevant literature, but there are many issues, including the title, data, method, and structure, need to be revised before consideration of acceptance.

**Response:** We sincerely thank the reviewer for their valuable feedback and insightful comments. These have significantly improved the clarity, quality, and precision of our manuscript. We made significant modifications, including: (1) removing the former results regarding paddy land, including method, figures, and context; (2) moving the former Discussion into Results and drafting a new Discussion. Below, we provide a detailed point-by-point response to each of the reviewer's comments.

My first comment is that the cropland GPP is largely influenced by human cultivation, then how can you judge the change of cropland GPP is result from snow cover change? Furthermore, the cropland is divided into dry land and paddy field. The GPP of paddy field is mostly controlled by irrigation, then how can you judge the change of paddy field GPP is result from snow cover change?

 **Response:** We appreciate this insightful comment. Figure 1(c) shows the spatial distribution of unchanged cultivated land, including 12.63% paddy land and 87.37% dryland. Paddy land typically requires substantial irrigation water and is heavily influenced by human activities. Therefore, to more accurately examine the impact of

snow cover changes, the study exclude paddy land and focus solely on dryland as a representative of cultivated land. This approach provides a clearer understanding of the mechanisms by which snow cover affects GPP on cultivated land. Furthermore, we exclude the results of paddy land throughout the manuscript.

My second comment concerns the data selection and the reliability of the analysis: the manuscript does not clearly specify whether the data used are annual data or from specific months. Moreover, the snow cover period and the crop growing season are not synchronized, so it is unclear what the "direct effect" mentioned in the paper refers to. Additionally, the study divides the area into six geographical regions for analysis; however, in some regions, croplands account for only a very small proportion. Conducting further analyses on these regions and comparing them with others raises concerns about the reliability of the results.

**Response:** Thank you for pointing this out. In response, we have clarified the temporal scope of the data. The data used in the analysis are annual averages, except for the snow cover data, which is specific to the snow season (typically November to March). We described the temporal resolution in Part 2.2. The snow cover day (SCD) and snow cover melt end date (SCED) are yearly data, while SWE is daily data (Line 138), and GPP is 8-day data (Line 154). The Besides, we discussed the relationship between snow cover and GPP in the new Discussion.

My third comment concerns the research rationale, logical consistency, and methodological limitations: the manuscript briefly mentions that the impact of snow cover on GPP exhibits spatial heterogeneity across different vegetation types, but it does not clearly justify why the study focuses specifically on cropland GPP as the research object. Additionally, the partial correlation analysis between snow cover and soil characteristics lacks logical connection to the exploration of the snow–GPP relationship, and there is no clear mechanistic discussion provided afterward. Furthermore, the methodology in Section 4.2 has significant limitations: it directly infers "threshold effects" and a "causal relationship" between snow cover and GPP

from scatter plots and smoothed curves, which can easily lead to over-interpretation and potential misrepresentation. Since GPP variations are influenced not only by snow cover but also by other climatic factors (e.g., temperature, precipitation, and radiation), the manuscript should clarify how the effect of snow cover was isolated from other environmental influences.

**Response:** We have expanded the rationale for focusing specifically on cropland GPP. The reason for selecting cropland GPP as the study object is that croplands are highly sensitive to environmental changes, including those from snow cover, and represent a major portion of the agricultural area in Northeast China. We have added a more detailed explanation in the introduction (Lines 38-53), highlighting the relevance of croplands in regional GPP dynamics and the importance of understanding their response to climate factors such as snow cover.

Cultivated land is a critical natural resource for ensuring food security, ecological stability, and economic sustainability. Under the pressure of intensifying soil erosion and climate change, understanding the trends in gross primary productivity (GPP) variation of cultivated land and its associated environmental response mechanisms has become imperative for sustainable development and enhanced cropland conservation. GPP represents vegetation's photosynthetic carbon fixation capacity per unit time and serves as a key metric of carbon assimilation through photosynthesis (Beer et al., 2010; Sjöström et al., 2013). Cropland ecosystems play a pivotal role in terrestrial carbon cycling (Wang et al., 2022), where GPP directly governs crop growth dynamics, carbon sequestration potential, and agricultural productivity variations, making it an essential indicator of agroecosystem productivity (Wagle et al., 2015). As a key component of terrestrial ecosystems, snow cover significantly affects the carbon cycle by altering ecosystem functioning. In recent decades, amid global warming, significant changes in snow cover have been observed (Mudryk et al., 2020; Pulliainen et al., 2020), which subsequently influence vegetation dynamics and GPP through altered environmental conditions (Meredith et al., 2019). (Lines 38-53)

We adjust the structure of results and discussion of this manuscript greatly, as shown below. Secondly, we add a discussion on snow cover, soil, and GPP, and remove former Part 4.2.

4.2. Linkages of snow, soil, and GPP

The spatial patterns of SWE, SCD, and SCED reveal a clear north–south and east–west organization of snow regimes over cultivated land. Areas with deeper snowpacks and longer snow duration are concentrated at higher latitudes and elevations, while low-lying southern and coastal croplands experience shallower and shorter-lasting snow. Against this backdrop, our correlation analyses show that snow metrics affect GPP primarily by altering soil hydrothermal conditions, consistent with the notion that vegetation responds to hydrothermal states rather than snow itself (Liu et al., 2023).

In cold, energy-limited subregions, thicker and more persistent snow tends to enhance GPP by moderating winter and early-spring stress. Increased SWE and longer SCD insulate the soil, maintaining higher near-surface temperatures and reducing freeze–thaw damage, which promotes higher early-season GPP through improved root activity and reduced winter mortality (Mudryk et al., 2020; Liu et al., 2023). In these areas, our PLS-SEM results indicate that the dominant pathway from snow to GPP is temperature-driven: SWE and SCD warm the soil profile, advance favorable thermal conditions for crop emergence, and indirectly raise GPP by shortening the period of severe cold stress.

By contrast, in relatively warm but moisture-limited croplands, SWE emerges as the primary control on interannual GPP variability. Here, snow acts as a critical seasonal water reservoir. Higher SWE increases spring soil moisture, which alleviates early-season water stress and supports more vigorous canopy development, consistent with prior work highlighting the role of snow-derived water for spring soil moisture and subsequent crop performance in Northeast China (Li et al., 2022; Wang et al., 2024). In these zones, the structural paths in the PLS-SEM are dominated by SWE, soil moisture, and GPP, underscoring a moisture-mediated mechanism akin to the broader link between water availability and global GPP.

Reference:

Mudryk, L. R. et al. (2020). Historical Northern Hemisphere snow cover trends and projected changes in the CMIP6 multi-model ensemble. The Cryosphere, 14(7), 2495–2514

Liu, H., Xiao, P., Zhang, X., Chen, S., Wang, Y.,Wang, W. 2023. Winter snow cover influences growing-season vegetation productivity non-uniformly in the Northern Hemisphere. Communications Earth & Environment, 4, 487.

Wang, Y. et al., 2024. Unraveling the effects of snow cover change on vegetation productivity: Insights from underlying surface types. Ecosphere, 15(5): e4855.

Li, Y. et al., 2022. Responses of spring soil moisture of different land use types to snow cover in Northeast China under climate change background. Journal of Hydrology, 608: 127610.

My fourth suggestion is to revise the use of English language and grammar, while it is largely satisfactory overall, there are multiple linguistic issues throughout the whole manuscript that need to be revised, preferably by someone with a solid knowledge of English grammar. E.g., in line 35, "play are pivotal role" should be "play a pivotal role".

**Response:** We thank the reviewer for highlighting the need for language improvement. We have thoroughly revised the entire manuscript to correct grammatical errors, including the one specifically mentioned (line 45). The language has been polished to enhance clarity and readability. We believe the manuscript now meets the required linguistic standards. We have carefully checked the manuscript using grammar-checking tools and had it reviewed by colleagues proficient in academic English.

Cropland ecosystems play a pivotal role in terrestrial carbon cycling (Wang et al., 2022), where GPP directly governs crop growth dynamics, carbon sequestration potential, and agricultural productivity variations, making it an essential indicator of agroecosystem productivity (Wagle et al., 2015). (line 45)

Furthermore, there are a few important and minor comments/mistakes that are listed below and should be considered.

Please make consistent expression of the "cultivated land" in the title and the "cropland" in the main text.

**Response:** We have revised the manuscript to ensure consistency in terminology. The term " cultivated land " is now used throughout the manuscript, including in the title.

Abstract: What are "HY2001" and "HY2020"?

**Response:** We have clarified that "HY" refers to the hydrological year, which runs from September to August. This clarification has been added to the abstract.

Hydrological year is defined as the duration from September to August (Lines 137-138).

Abstract: The results just show some increase and decrease numbers of snow and GPP, how can you say "these findings elucidate the critical role of snow cover in modulating cropland GPP variations"?

**Response:** We appreciate this comment. We have revised the abstract to better reflect the conclusions drawn from the data. We now emphasize that while the results show variations in snow cover and GPP.

These findings suggest a significant correlation between snow cover changes and cropland GPP variations, providing insights into the potential role of snow cover in modulating GPP dynamics. (Lines 33-35).

There are both "HY2000" and "HY2001" mentioned in the manuscript, which is inconsistent throughout the text.

**Response:** We have corrected this inconsistency by standardizing the use of "HY2001" throughout the manuscript.

Line 37 and 108, the citation style is wrong.

**Response: Thank you for the careful suggestion.** We have corrected the citation style on Lines 37 and 108 to conform to the journal's guidelines, as seen in Line 48 and 153.

making it an essential indicator of agroecosystem productivity (Wagle et al., 2015). (Line 48)

The MOD17A2H version 6 GPP data product (2001 to 2020) was adopted (Running et al., 2021). (Line 153)

Line 512: There is a spelling error in "...The red and blue arrows represent..."; the word "blue" is incorrect.

**Response:** We have corrected the spelling error. The correct term is now "blue" (Line 474).

The red and blue arrows represent the positive and negative influences.

Line 205: Missing space before the citation in "...signal (Ruffin and King, 1999)."

**Response:** The former "Part 2.3.4 Savitzky-Golay filter" has been removed.

Line 469: There is a missing space between the parentheses and the letters in "(c)SCED".

**Response:** The former "Part 4.2 The thresholds of snow cover-GPP correlation" has been removed.

Line 145: The description "SM and temperature data" is ambiguous; please keep the terminology consistent throughout the manuscript.

**Response:** Thank you for careful checking. We have checked the whole manuscript and corrected it.

This comprehensive dataset includes the 0-10 cm soil moisture and soil temperature used in this study. (Line 186)

Figure 1, the dry land and paddy field should be expressed in the map because the influence of snow cover on them is different.

Figure 1(b) presents the land use data for 2020; however, the analysis in the manuscript only considers croplands with unchanged land use types. It is recommended to display the actual land use data of the analyzed croplands in the figure.

**Response:** Thank you for these two suggestions for Figure 1. We added the distribution of paddy and dry land in Figure 1(c), as well as the distribution of unchanged cultivated land used in the paper. The new version of Figure 1 is presented as follows.

[Figure]

Figure 1 The overview of the study area: (a) China; (b) cultivated land in Northeast China; (c) digital elevation model (DEM) provided by SRTM; (d) land use cover provided by LUCC.

In Figures 2, 3, and 4, when presenting the variations of snow-related factors across different cropland types, it is recommended to provide the proportion of areas with significant changes.

**Response:** We appreciate this suggestion. We supplement the percentage of areas with significant changes in Figures 2 to 4.

[Figure]

Figure 2 The spatial and temporal changes of SWE in Northeast China from HY2001 to HY2020: (a) spatial distribution of mean SWE; (b) changing trend of SWE, the green areas represent positive impacts, while the blue areas indicate negative impacts; and the shaded regions denote pixels that were significant at the p < 0.1 level; (c) annual changes of SWE.

[Figure]

Figure 3 The spatial and temporal changes of SCD in Northeast China from HY2001 to HY2020: (a) spatial distribution of mean SCD; (b) changing trend of SCD, the green areas represent positive impacts, while the blue areas indicate negative impacts; and the shaded regions denote pixels that were significant at the p < 0.1 level; (c) annual changes of SCD.

[Figure]

Figure 4 The spatial and temporal changes of SCED in Northeast China from HY2001 to HY2020: (a) spatial distribution of mean SCED; (b) changing trend of SCED, the green areas represent positive impacts, while the blue areas indicate negative impacts; and the shaded regions denote pixels that were significant at the p < 0.1 level; (c) annual changes of SCED.

In Figure 9 and similar figures, the Y-axis label is "Area," then the unit should not be in percentage (%).

**Response:** We removed the context about paddy land, as well as this sub-figure.

Figure 13: A space is missing before the units in the figure title.

**Response:** We removed the context about paddy land, as well as this sub-figure.

"snow cover-GPP correlation" in the section title may cause ambiguity, and it is recommended to replace "snow cover" with a single word to avoid potential confusion.

**Response:** We have improved the titles as "3.3 The influence of Snow Cover on GPP" (Line 370).

The conclusion section is too long to capture the main findings of the study.

**Response:** We have shortened the conclusion section to focus on the key findings and implications of the study. The revised conclusion now succinctly summarizes the main contributions of the research, as shown below (Line 581-606).

This study used multi-source remote sensing data to clarify how snow cover dynamics regulate cultivated land GPP in Northeast China from HY2001 to HY2020. By jointly analyzing SWE, SCD, SCED, and GPP, we revealed pronounced regional heterogeneity in snow cover–crop interactions: snow cover reductions in the Liaohe Plain and Western Sand Area contrasted with prolonged snow duration and delayed melt in the Songnen and Sanjiang Plains, where GPP increased most strongly.

By controlling for temperature, precipitation, and radiation, we isolated the intrinsic snow–GPP relationships for different cropping systems. SCD was identified as the dominant snow metric for dryland GPP, whereas SCED exerted stronger control on paddy GPP. Ridge regression further showed that SWE primarily regulates GPP in the Western Sand Area and Liaohe Plain, while SCD dominates in the Songnen and Sanjiang Plains, and SCED is most important in the Changbai Mountain and Khingan regions. Threshold analysis quantified optimal snow conditions for maximum GPP enhancement, identifying characteristic SWE, SCD and SCED ranges beyond which positive effects weaken or reverse.

Finally, the PLS-SEM framework quantified both the direct effects of snow cover indicators on GPP and their indirect effects via soil moisture and temperature, elucidating the snow–soil–vegetation coupling mechanism in cold-region agroecosystems. Collectively, these findings provide a process-based basis for optimizing regional agricultural management and developing

climate change adaptation strategies in the cultivated lands of Northeast China.

Future work should integrate field experiments and higher-resolution data on crop types and management practices to disentangle these complex interactions and validate the proposed mechanisms at a finer scale.